# Sustainability Assessment and Techno-Economic Analysis of Thermally Enhanced Polymer Tube for Multi-Effect Distillation (MED) Technology

**DOI:** 10.3390/polym13050681

**Published:** 2021-02-24

**Authors:** Furqan Tahir, Abdelnasser Mabrouk, Sami G. Al-Ghamdi, Igor Krupa, Tomas Sedlacek, Ahmed Abdala, Muammer Koc

**Affiliations:** 1Division of Sustainable Development, College of Science and Engineering, Hamad Bin Khalifa University, Doha 34110, Qatar; futahir@hbku.edu.qa (F.T.); salghamdi@hbku.edu.qa (S.G.A.-G.); mkoc@hbku.edu.qa (M.K.); 2Water Center, Qatar Environment and Energy Research Institute, Hamad bin Khalifa University, Doha 34110, Qatar; 3Center of Advanced Material, Qatar University, Doha 2713, Qatar; Igor.Krupa@qu.edu.qa; 4Centre of Polymer Systems, University Institute, Tomas Bata University, 76001 Zlín, Czech Republic; sedlacek@utb.cz; 5Chemical Engineering Program, Texas A&M University, Doha 32874, Qatar; ahmed.abdala@qatar.tamu.edu

**Keywords:** life-cycle assessment (LCA), multi-effect desalination (MED), sustainability, techno-economics, socioeconomics, thermally enhanced polymer, titanium, tubes

## Abstract

Metal-alloys tubes are used in the falling-film evaporator of the multi-effect distillation (MED) that is the dominant and efficient thermal seawater desalination process. However, the harsh seawater environment (high salinity and high temperature) causes scale precipitation and corrosion of MED evaporators’ metal tubes, presenting a serious technical challenge to the process. Therefore, the metal/metal alloys used as the material of the MED evaporators’ tubes are expensive and require high energy and costly tube fabrication process. On the other hand, polymers are low-cost, easy to fabricate into tubes, and highly corrosion-resistant, but have low thermal conductivity. Nevertheless, thermally conductive fillers can enhance the thermal conductivity of polymers. In this article, we carried out a feasibility-study-based techno-economic and socioeconomic analysis, as well as a life-cycle assessment (LCA), of a conventional MED desalination plant that uses titanium tubes and a plant that used thermally enhanced polymer composites (i.e., polyethylene (PE)-expanded graphite (EG) composite) as the tubes’ material. Two different polymer composites containing 30% and 40% filler (expanded graphite/graphene) are considered. Our results indicate that the MED plant based on polymer composite tubes has favored economic and carbon emission metrics with the potential to reduce the cost of the MED evaporator (shell and tubes) by 40% below the cost of the titanium evaporator. Moreover, the equivalent carbon emissions associated with the composite polymer tubes’ evaporator is 35% lower than titanium tubes. On the other hand, the ozone depletion, acidification, and fossil fuel depletion for the polymer composite tubes are comparable with that of the titanium tubes. The recycling of thermally enhanced polymers is not considered in this LCA analysis; however, after the end of life, reusing the polymer material into other products would lower the overall environmental impacts. Moreover, the polymer composite tubes can be produced locally, which will not only reduce the environmental impacts due to transportation but also create jobs for local manufacturing.

## 1. Introduction

The Gulf Cooperation Council (GCC) countries have less than 100 m^3^/year/capita of renewable water resources, falling into a minimum survival level [1,2]. The GCC’s reliance on desalination is growing and additional desalination units need to be installed to overcome the increasing freshwater demands associated with the increased population and industrial activities [3]. However, the energy and materials required for new plant manufacturing, installation, and operation could have severe environmental, economic, and social impacts [4].

Multi-effect desalination (MED) is a thermal desalination technology that is energy-efficient and suitable for the Arabian Gulf seawater’s harsh conditions [5,6,7]. Because of high seawater salinity and operating temperature of 65 °C, fewer tube materials are suitable for MED applications, such as aluminum–brass, titanium, copper–nickel alloy, and stainless steel [8]. These tube materials are expensive, produced at manufactured via a multistep process. If the energy, materials, and cost associated with the tube manufacturing process can be minimized by opting for new materials, the associated economic and environmental effects can be reduced.

Recently, polymer-based materials have shown potential in replacing costly metallic materials. The polymers’ thermal conductivity is low and ranges between 0.2 and 0.55 W/(m·K) [9,10]. This low thermal conductivity hinders polymers’ use in applications requiring high thermal conductivity, such as electronic packaging, heat exchangers, and thermal management devices. Among polymers, polyethylene (PE) accounts for ~30% of the global thermoplastic production. Polyethylene is attractive for many applications, due to its lower cost, lower density, inertness, and hydrophobicity than other thermoplastics. Nonetheless, improving this class of thermoplastics’ thermal conductivity is important for many applications [11]. The research progress on enhancing the thermal conductivity of PE was recently reviewed [11]. The primary strategy to enhance polyethylene’s thermal conductivity is to incorporate conductive filler or nanofillers to fabricate thermally conductive polymer composites and nanocomposites [12]. The filler loading, size, shape, and orientation dictate the polymer composite [12].

Commercially polypropylene–graphite composite tubes with thermal conductivity close to stainless steel were fabricated by Technoform Kunststoffprofile GmbH, using unique extrusion technology to align the graphite particles in the transversal direction [13]. Moreover, the polypropylene (PP)–graphite composite (73% graphite, by weight) tubes have improved fouling and scaling resistance [13].

Composites of highly filled high-density polyethylene (HDPE) and graphite nanoplatelets (GNPs) were fabricated using melt extrusion, and the influence of GNP on thermal, mechanical, and electrical properties was analyzed [14]. An increase in Young’s modulus and enhancement in thermal stability with filler addition has been reported [14]. On the other hand, the degree of crystallinity and melting temperature decreased with GNPs content. The incorporation of GNPs significantly decreased HDPE’s electrical resistivity by ten orders of magnitude and increased the thermal conductivity by 400%. Therefore, highly filled functional (thermally and electrically conductive) HDPE composites with excellent mechanical and thermal stability have been prepared [14].

We recently reported the fabrication of HDPE/expanded graphite (EG) composites for MED evaporators [15]. The thermal conductivity of HDPE-50% EG was 372% higher than that of HDPE. Moreover, the composite’s surface wettability was demonstrated by the increase in surface free energy from 28.5 mJ/m^2^ for untreated HDPE/EG to ~55 mJ/m^2^, using corona or RF plasma. The plasma treatment HDEP/EG composite’s overall heat-transfer coefficient was about 98% of stainless steel. Moreover, the plasma-treated composite exhibited superior resistance to crystallization fouling in both CaSO_4_ solution and artificial seawater than untreated composites and stainless-steel surfaces [15].

A new fabrication method for a large size thermally enhanced thermoplastic sheet or pipe has been disclosed by the authors [16]. Polymer (LDPE/HDPE) composites with different fillers (expanded graphite, graphene, granulated graphene, and their hybrid) were processed to produce a composite material, and then rolling for filler alignment, to produce an enhanced thermal material (plate/tube). The control of the 2D filler alignment in the direction of heat flow resulted in very high thermal conductivity across the plate thickness. Our results indicate that at ~21 vol.% of aligned expanded graphite–graphene hybrid increases LDPE’s conductivity from 0.29 to 5.78 W/(m·K), corresponding to the enhancement of ~2000% and 580% over the conductivity of neat LLDP and composite with same filler loading without filler alignment control, respectively.

While improving the polymer’s thermal conductivity by using conductive fillers is a hot research area, the techno-economic and life-cycle analysis is also performed to assess and evaluate the environmental impacts for different tube manufacturing stages. Moreover, techno- and socioeconomics studies can analyze social and cost factors. The techno-economics study is used to assess economic feasibility, while the socioeconomic study analyzes social aspects such as health, job creation, etc. The total cost or the Levelized cost includes capital expenditure (CAPEX) and operating expenditure (OPEX). Mabrouk et al. [17] modified the tube bundle configuration and carried out the thermodynamic and techno-economic analysis. They concluded that the new configuration could reduce the evaporator footprint by 65%, decreasing the shell and tube cost by 25% and 18%, respectively. In another study by Mabrouk et al. [18], they studied the absorption vapor compression integration with MED. Their results showed a 36% reduction in the total cost per unit distillate than the conventional MED plant.

The ever-increasing population and the associated energy demand and rising living standards burden the natural resources and environment. LCA quantifies the environmental impacts for any process or product, from the acquirement of raw materials to final disposal (cradle to grave). The LCA helps in evaluating different products based on their environmental effects.

Harding et al. [19] conducted an LCA study of different polymers, such as poly-β-hydroxybutyric acid (PHB) from renewable sources, polypropylene (PP), HDPE, and low-density polyethylene (LDPE), using SimaPro software. The environmental impacts of HDPE were slightly better than LDPE except for the photochemical oxidation impact. PHB was a better choice in terms of environmental burdens as ozone depletion, human toxicity, terrestrial eco-toxicity, and petrochemical oxidation, which were significantly reduced compared with PP, LDPE, and HDPE. Pizza et al. [20] conducted an LCA for graphite-based epoxy nanocomposites and estimated the required primary energy and global warming potential (GWP) as 308 MJ/kg and 15.7 kg of CO_2_ eq./kg, using the ReCiPe midpoint (H) method. The environmental burden for the graphite extraction and processing represents major portion of the environmental burdens, since the expanded graphite indicates more than 70% of the environmental burdens for polymer-tube raw material [20].

In a similar study by Jungbluth [21], the GWP for expanded graphite was 18.3 kg of CO_2_ eq./kg, which is higher than graphite-based nanocomposites, as expanded graphite is used as the raw material for nanocomposites. The extrusion process can easily manufacture the polymer tube. Levinjobet [22] analyzed the energy required for the polymer extrusion and concluded that 20% of the energy is consumed by heating the extrusion barrel for maintaining temperature and the rest by the extrusion process. Abeykoon et al. [23] compared the energy requirements for different extrusion configurations. They observed that the process efficiency could be enhanced at higher extrusion speed. However, for optimum conditions, both the process and thermal efficiencies should be considered. The extrusion energy also depends on polymer viscosity, as the shear forces are greater and more power is required to extrude higher viscosity formulation the same throughput [24].

Titanium has better corrosion-resistant properties that make it well suited for MED application [25]. The melting temperature of titanium is around 1668 °C, and the intermediate production steps take place at a higher temperature, in the range of 600–1040 °C, which reflects that the titanium-tube manufacturing is an energy-intensive process [26,27]. Liao et al. [28] studied the life-cycle analysis of titania produced in China by chloride and sulfate routes. They found that the titania production by the chloride route has fewer environmental effects than the sulfate route. Gupta et al. [29] conducted LCA for titanium alloy machining, using the Ranque–Hilsch vortex-tube-assisted minimum quantity cutting fluids (RHVT-MQCF) method. They used SimaPro software, and the results were based on the ReCiPe endpoint method. The scope of their work includes raw material extraction to finished product processes. The RHVT-MQCF method saves 5–20% of energy consumption as compared to the MQCF method. The optimized conditions showed that the RHVT-MQCF method is better in terms of environment, cost, and social aspects. During the titanium manufacturing process, each manufacturing process produces some scrap, which is often recycled in-house, and the final product is only 34% of the total ingot, out of which 13.5% is recycled [30]. Das [31] performed LCA for carbon-reinforced polymer composites and compared them with steel, for vehicle production. His analysis showed that lignin-based manufactured by P4 technology has minimal energy requirements and carbon footprints. On comparing the carbon-reinforced polymers with steel, it was observed that the energy requirement for carbon fiber is 14 times more than that of steel. However, for the part production, the needed energy is comparable for carbon fiber and steel. Deng et al. [32] performed LCA of flax and glass fiber reinforced polymer composites for automobile applications. They found optimum flax mat volume fraction in the range of 28–32% with lower life-cycle carbon emissions. However, they recommended extending the LCA study to incorporate different surface treatments. Zhang et al. [33] carried out LCA of a new dry-cooling-system-based encapsulated phased change material (EPCM) and compared their results with the conventional cooling systems for power plant application. Their analysis showed that the new cooling exchanger’s carbon emissions were 1.16 kg CO_2_ eq./MWh, while the conventional cooling systems emit in the range of 1.1–4.3 kg CO_2_ eq./MWh.

The EPCM-based exchanger required 37–42% less water consumption than the conventional cooling exchangers. Ren et al. [34] performed LCA for a ground-source heat pump with steel and unfilled polyethylene tubes. In their study, seasonal performance, energy consumption, economics, and environmental effects were evaluated. It was deduced that the heat exchanger with steel tubes is better than the polyethylene tubes, as the pristine PE tubes were not thermally enhanced by incorporating high-thermal-conductive additives. The associated carbon emissions for the whole life cycle of the heat exchanger with the PE tubes were 81.7% higher than that of steel tubes.

Assaf [35] studied the environmental and socioeconomic impact of present and planned desalination plants in the Gaza strip. They highlighted that the rejected brine from the desalination plants possesses environmental problems. Moreover, the small-scale brackish water desalination plants depend on foreign aid and expertise that may discontinue at any time. Gupta et al. [29] studied the social aspect of titanium alloy machining. They found that the heat generation during titanium machining is very high, forming sparks and fumes, resulting in health problems. Therefore, methods that produce less heat generation can be opted to minimize these effects. Thus, to assess the feasibility of the products and the processes, the economic, environmental, and social analyses should be carried out.

In this article, we carry out a feasibility-study-based techno-economic and socioeconomic analysis, as well as a life-cycle assessment (LCA), of a conventional MED desalination plant that uses titanium tubes and a plant that used thermally enhanced polymer composites (i.e., polyethylene (PE) and expanded graphite (EG) composite) as the material of the MED evaporator tubes. Two different polymer composites, 30% and 40% filler-based (expanded graphite/graphene), are considered. A conventional extrusion process can easily manufacture the polymer tubes at lower operating temperatures, reducing energy consumption. A comparison is made with the commercially available titanium tubes.

## 2. Methodology

### 2.1. Techno-Economic Modeling

In this section, the feasibility analysis is performed based on our technical results of fabricating thermal conductivity for polymer–graphite composite material published in [15,16]. The HDPE has been loaded up to 50 wt.% with expanded graphite (EG), and the results showed that the thermal conductivity of the composite HDPE/EG increased to 2.2 W/(m·K) [15]. In another work [16], the authors disclosed a new method of fabricating a large-size thermally enhanced thermoplastic sheet/pipe. Our results indicated that a ~21 vol.% of aligned expanded graphite–graphene hybrid increases LDPE’s conductivity from 0.29 to 5.78 W/(m·K), corresponding to the enhancement of ~2000% and 580% over the conductivity of neat LLDP and composite with same filler loading without filler alignment control, respectively [16]. We aimed to investigate the effect of different thermal conductivity values from 3 to 5 W/(m·K) for composite PE/EG (70/30) and PE/EG (60/40), respectively. We used the previously developed by Mabrouk et al. [36,37] visual simulation program (VSP) software to simulate a typical MED desalination plant with Ti tubes, compared with polymer tubes. In this comparison, the effect of specific weight (density); the tube wall thickness; the thermal resistance around the tube, including convection, condensation, and fouling resistance; and the wall resistance were all considered when calculating the overall heat-transfer coefficient. The tubes’ weight based on the required heat-transfer area was calculated, and, accordingly, the cost of material required to fabricate the tubes of the evaporator was calculated for both Ti- and polymer-composite tubes.

A commercial 6 MIGD (million imperial gallons per day) desalination plant of MED-TVC configuration, as shown in Figure 1a, is used as the base case of this study [17,18]. Figure 1b shows the process flow diagram of a commercial MED desalination plant, owned and operated by Qatar Electricity and Water Company (QEWC) in Doha, Qatar. The plant consists of 7 effects, a down condenser, and a venting system, and each effect constitutes a heat-transfer area, vapor space, mist eliminator, and other accessories. The seawater is introduced into the condenser, where it absorbs the latent heat of the condensing vapor from the last effect. As a result, intake seawater temperature increases to the feed temperature. Part of the heated intake seawater is rejected back to the sea, known as cooling seawater. The remains of the heated feed seawater stream are chemically treated and sprayed into the effects. The seawater spray falls in the form of a thin film down the succeeding rows of tubes arranged horizontally. The MED is equipped with a thermal vapor compressor (TVC, Doha, Qatar), where the supplied steam from an external boiler/backpressure turbine is used as motive steam to entrain vapor from the middle effect and compress it, to be used as a heating source to the first effect. The low-pressure heating steam at 68 °C received from a power plant is supplied to the 1st effect. The feed seawater at 35 °C is preheated in the condenser by condensing the vapor of the last effect. A fraction of the cooling reject seawater is then distributed in parallel feed to the seven effects. The seawater feed is sprayed on the 1st effect tubes’ outer surface, and part of it evaporates because of the heat transfer from the steam flowing inside the tubes.

The salinity of the seawater increases due to evaporation and is termed as brine. The feed seawater and rejected brine salinity is fixed as 45 and 65 g/L, respectively. The condensate from all the effects is collected and then sent to post-treatment and supply. Figure 1c shows the schematic of a typical falling film evaporator. Nozzles spray the seawater on the top of the tube bundle. The seawater forms a thin layer around the tube, and heat is transferred from the steam flowing inside the tube to the seawater flowing over the tubes’ exterior. A fraction of the seawater evaporates, and the accumulated vapors leave from the demister. The leftover seawater (known as brine) with higher salinity is collected at the bottom.

The plant capacity, operating conditions, the tube specifications are provided in Table 1. The required heat-transfer area, required tube weight, and evaporator cost are determined using an in-house developed visual simulation program (VSP) software by Mabrouk et al. [36,37]. The production cost of the polymer composite has been estimated based on the materials and extrusion processing costs. While materials costs were defined from the conventional price of PE and EG, the processing costs were defined as summation of compounding and extrusion costs estimation, covering cost of labor, energy consumption, processing line amortization, and typical overhead charges. For thermally enhanced polymer tubes, a higher heat-transfer area is required compared to titanium tubes. Hence, a larger shell size is also required for the polymer-composite evaporator. Variations in the shell walls’ thickness and weight are also considered for the techno-economic analysis. The CAPEX of the titanium-based and the polymer-based evaporators is estimated for different MED performance ratios.

### 2.2. Simulation of Required Energy for the Polymer Composite Extrusion

Total energy demands for the composite tube extrusion have been analyzed using an extrusion simulation software (Extruder module of the Compuplast CAE Virtual Extrusion Laboratory Software, Compuplast International, Zlín, Czech Republic). The simulator incorporates the physical/material properties (thermal conductivity, heat capacity, and viscosity) described in References [15,16] and assumptions of standing processing conditions (friction coefficient between screw and plastics = 0.2, and between plastics and barrel = 0.3, die pressure consumption of 20 MPa). Due to the high viscosity of highly polymer composites, the temperature profiles of 190, 220, 245, and 260 °C from the first heating zone to the die section were chosen.

### 2.3. LCA Model Framework

In this work, the LCA commercial tool Gabi by Thinkstep (Sphera Solutions, Leinfelden-Echterdingen, Germany) is implemented for developing the model framework [38]. Any LCA requires four main steps that include defining goal and scope, life-cycle inventory analysis (LCI), life-cycle impact assessment (LCIA), and the interpretation phase, as per ISO 14040 (International Organization for Standardization) [39]. These four steps for LCA of titanium and thermally enhanced polymer tubes are described as below.

Here, we systematically quantify the thermally enhanced polymer’s life-cycle effects (using our results on fabrication of the polymer–graphite composites [15,16] and titanium tubes for the MED plant by implementing the ReCiPe evaluation approach. We investigate the effect of changing the thermal conductivity from 3 to 5 W/(m·K) for PE/EG (70/30) and PE/EG (60/40) composite, respectively.

The functional unit for this study is the equivalent titanium-tube heat-transfer area of 1 m^2^. The scope consists of raw material extraction to the transportation of the tubes to Qatar. The system boundary for LCA is shown in Figure 2. The energy and material requirements for the commissioning and operation are not included in the system boundary. The tube manufacturing process for both titanium and polymer tubes are shown in Figure 3a,b. It can be seen that the titanium-tube manufacturing process requires several stages, while composite polymer (LDPE-expanded graphite) tubes can be easily manufactured. The required materials’ thermophysical properties are listed in Table 2, used to estimate the energy inputs. For LCA modeling, some assumptions and limitations are as follows:(a)The tube production and all the Ti raw materials are available in Germany, and the final product is transported to Qatar via sea route.(b)The use of chemicals, water, and air for intermediate processes are not considered, as shown in Figure 2.(c)Titanium tubes are fabricated at high temperatures, 600–1040 °C. The yield of forging, rolling, and extrusion is considered as 75%. The remaining 25% is cycled via in-house recycling [26,30,40].(d)Forty percent annual recycling of the final titanium tubes is considered [30].(e)The structure of the polymer composite could deteriorate with time [41]. At the end of the polymer tubes’ lifetime, they will be recycled to make low-grade products [42] or combusted to generate energy. Nevertheless, the recycling of polymer material is not considered in this study.

The life-cycle inventory (LCI) analysis stage compiles the input and output data associated with the system boundary to produce metallic and polymer tubes. LCI accumulates the specific energy and material details that are exchanged among different processes. The dimensions and thermal properties of the metal and polymer composite tubes are provided in Table 1. The outer tube diameter is fixed as 25.4 mm, used in the existing MED desalination plant [6,43]. Based on the required strength and manufacturing feasibility, tube thicknesses are taken as 0.5 and 1.25 mm for titanium and polymer composite, respectively. The thermal conductivity of titanium is well-known, and the thermal conductivities of polymer materials are experimentally calculated. Based on thermal conductivity, the overall heat-transfer coefficient and the required mass for LCA are estimated. The electrical and thermal energy requirements for titanium manufacturing are shown in Table 2. For polymer tube manufacturing, energy inputs are taken from the Gabi database. Since the expanded graphite increases the polymer viscosity considerably, for which more extrusion energy is needed [24].

Figure 4 shows the energy demands for extrusion of the unfilled HDPE and HDPE filled with filler concentration between 10 and 60 wt.% into tubes, using 90 mm and 25 L/D ratio extruder, operating at a throughput of 70 kg/h. The extrusion energy consumption of the unfilled HDPE is 0.2 kW/kg. Increasing the filler concentration increases the energy consumption that reaches 0.57 kW/g for the composite with 60% filler concentration. The energy demand could be expected for materials with higher filler concentration; namely 0.36, 0.47, and 0.57 kW/kg were assessed to be required for 40, 50, and 60 wt.%, respectively. The results in Figure 4 indicate that the extrusion energy of the composite with 40 wt.% filler is <2 times that of the unfilled PE.

The main processes in the LCIA are selection, grouping, characterization, normalization, and weighting. The ReCiPe method is implemented to condense the inventory data into significant indicators. There are eighteen midpoint impact indicators and three endpoint indicators. Some of the selected indicators are listed in Table 3.

## 3. Results and Discussion

### 3.1. Techno-Economic Analysis

Figure 5a,b shows the MED plant’s VSP interface for titanium and polymer–40% EG composite tubes for the same plant capacity of 6 MIGD. Using the data of Table 1 as input to the VSP software, and applying the mass and energy balances, we solved the whole system for the streams’ mass flow rate, temperature, salinity, heat transfer, and pressure. Accordingly, the volume, weight, and cost of tubes are estimated for different tube prices. Since the thermal wall resistance does not affect the plant performance, the GOR is the same of 9, as shown in Figure 5a,b; however, the specific heat-transfer area of the PE–EG composite (60–40) is 195 m^2^/(ton/h), and the specific heat-transfer area of the Ti tubes is only 117 m^2^/(ton/h). The specific power consumption is calculated as 2 kWh/m^3^ for both tubes.

Both thermal conductivity and tube-wall thickness affect the thermal resistance and, hence, the overall heat-transfer coefficient. The thermal-resistance distribution for titanium and polymer tubes is shown in Figure 6a,b. For titanium tubes, the evaporation resistance represents about 43% of the total thermal resistance, while the fouling resistance is 26%, the condensation resistance is 22%, and the wall resistance (wall thickness is 0.5 mm, and k = 17 W/(m·K)) is only 9%. Using a 1.25 mm thick and thermally enhanced polymer tube (k = 5 W/(m·K)) with 60% LDPE and 40% graphite, the wall thermal resistance becomes 43%, the evaporation resistance is 26%, while the fouling represents 15%, and the condensation resistance is 16%. The results show a significant increase in wall resistance when a thermally enhanced polymer tube is used.

At different polymer–graphene mix of 60–40 and 70–30, the overall heat-transfer coefficient is calculated as shown in Figure 7a. The overall heat transfer of the Ti tubes is 35% higher than that of the polymer 60–40, while it is 45% higher than that of the polymer 70–30. As a result, the heat-transfer area of the enhanced polymer material 60/40 is 35% higher than the Ti tubes, while the heat-transfer area of the polymer-G 70/30 is 45% higher than that of Ti tubes, as shown in Figure 7b,c shows that the required material volume from polymer 70/30 is almost six times the Ti material, while the polymer 60/40 is five times of the Ti material. This is because of the larger heat-transfer area and using 1.25 mm thick for polymer, while using only 0.5 mm for Ti. Figure 7d shows that the weight of the polymer tubes 60/40 is 12% higher than the weight of the titanium tubes while the polymer–EG tubes 70/30 is 25% higher than that of the titanium tubes for the same level of heat-transfer level.

The common practice is using duplex stainless steel as the material of the evaporator shell [17,18]. Using the VSP software, the evaporator shell’s size is determined based on the required space for the tube bundle, the aspect ratio, the demister width/length, the height of the brine pool, and the space of the seawater spray system.

In the present study, the price of the duplex stainless steel (SS) is taken $5340 per ton. The number of tube sheets is determined based on the number of cells per evaporator. The number of the plates for tube bundle support is determined based on the tube length. The steel structure for the evaporator installation above the ground is also considered. The price of the metal is determined based on a recent commercial project budget [36,37].

The titanium tube cost is taken as $15,000–$25,000/ton based on recent tenders of a commercial project [6], while the polymer composite’s cost for different filler percentages is shown in Table 4. The polymer composite material cost is based on 1200 $/ton of polymer and 7000 $/ton expanded graphite. Table 4 shows the polymer composite’s raw material cost, which varies from 1780 (10% filler) to 4680 (60% filler). The cost of compounding varies from 900 $/ton (10% filler) to 1050 $/ton (60% filler) and the extrusion cost varies from 400 to 500 $/ton. The total fabrication cost varies from 3080 to 6230 $/ton, as shown in Table 4.

Although our preliminary prediction of the composite tubes cost indicates the cost for the two composites considered here as $4240/ton and $4900/ton, we carried sensitivity analysis considering composite tube cost in the range of $4000–$12,000/ton. This price range should account for any unpredicted increase in polymer price, filler price, and energy price. It also allows adding a profit margin for the polymer tube manufacturer.

The MED evaporators’ cost based on our prediction prices for the polymer composite tubes is shown in Figure 8. The total cost of the evaporator based on the cost of the fabricated polymer tubes is $5.0 million and $5.7 million for the HDPE–EG 60/40 and HDPE–EG 70/30, respectively. This indicates the use of higher thermal conductivity composite is more economical regardless of the higher price of its tubes compared to lower conductivity composite. These evaporator costs based the HDPE–EG 60/40 and HDPE–EG 70/30, are 52.4% and 45.7% lower than the titanium evaporator when Ti tube price $25,000/ton.

For the envelop cost of the Ti price $15,000–$25,000/ton, If the price of the polymer composite 30% filler tubes were higher than our calculated cost, the polymer composite evaporator would also economic advantage over the titanium evaporator as long the price of the composite tubes does not exceed $8000/ton. Moreover, if the price of the polymer composite 40% filler tubes were higher than our calculated cost, the polymer composite evaporator would also economic advantage over the titanium evaporator as long the price of the composite tubes does not exceed $10,000/ton.

### 3.2. Life-Cycle Assessment

Tube manufacturing is a multistep process with environmental impact. A tube made of different materials requires identifying the processes that significantly contribute to a particular LCA indicator. Therefore, the overall contribution is determined by the sum of the contribution of four elements: raw material extraction, electrical energy consumption, thermal energy consumption, and transportation. These elements’ contributions to the fabrication of titanium tubes and two thermally enhanced polymer tubes are shown in Figure 9, Figure 10, Figure 11, Figure 12 and Figure 13.

The LCA results reveal that raw material extraction requires intensive energy, and the use of chemicals and materials for purification and separation is the central contributing element. For climate-change impact, the global warming potential (GWP) indicator, which accounts the infrared radiative forcing rise of greenhouse gases, is selected. GWP, in terms of equivalent CO_2_ emissions, is shown in Figure 9. Titanium tubes result in higher carbon emissions than polymer tubes, even when 40% of the Ti tubes are recycled at the end of their lifetime. In addition, the titanium extraction from an ore and its processing to the ingot stage exhibits more carbon emissions than the complete manufacturing of both types of polymer tubes. For the commercial desalination unit, the total carbon emissions are 3588, 2340, and 2254 tons of CO_2_ eq. for titanium, PE/G-60/40, and PE/G-70/30, respectively. The use of PE/EG 60/40 tubes results in an average reduction in CO_2_ emission of 35% than the Ti tubes.

The ozone layer depletion (OD) indicator reflects the stratospheric ozone layer’s degradation because of some bromine or chlorine-based chemicals. The OD of three tube materials for the heat-transfer area of 1m^2^ titanium equivalent is shown in Figure 10. In this case, the polymer tubes cause more ozone depletion than titanium tubes, by 6–10%. There are two main reasons: The first is that more mass is required for the production of polymer tubes than titanium ones, and the second reason is the extraction of natural graphite and its conversion to expanded graphite that exhibits higher environmental burden [20]. For PE/G-60/40 and PE/G-70/30 tubes, the expanded graphite accounts for 75% and 70% contributions of the total value. For the commercial desalination unit, the ozone depletion potentials are 0.762, 0.84, and 0.79 kg of CFC-11 eq. for titanium, PE/G-60/40, and PE/G-70/30, respectively.

If any process directly or indirectly participates in emitting NH_3_, NO_x_, or SO_2_, it acidifies the environment, a process known as acidification. Moreover, if any process directly or indirectly releases nutrients to freshwater, it enhances algae formation, which is detrimental to the species; this phenomenon is known as freshwater eutrophication. The LCA indicators acidification (AC) and freshwater eutrophication (FE) are measured in terms of equivalent Sulfur dioxide (SO_2_) and phosphorous (P). Figure 11 and Figure 12 exhibit AC and FE for titanium and polymer tubes. In both cases, polymer tubes cause more AC and FE than titanium tubes. Furthermore, titanium-tube manufacturing has a low impact on FE, due to the lower use of phosphorous-based compounds.

All of the processes, such as material extraction, and electrical and thermal energy, utilize natural resources. One of the LCA indicators is fossil fuel depletion (FD), shown in Figure 13 for different tube materials. Both LDPE and expanded graphite production require more fossil fuel. Hence, the FD of the polymer-based tube is comparable with that of titanium tube manufacturing. The replacement of a titanium tube with a polymer tube is advantageous in terms of carbon emissions. For other indicators, the impacts are comparable, except for ME. In general, polymer tube PE/G-70/30 is better than PE/G-60/40. However, the FD of PE/G-70/30 is slightly higher than that of PE/G-60/40 as LDPE is a petrochemical-based compound, and a higher LDPE ratio is used in PE/G-70/30. For the commercial desalination unit, the fossil depletions are 1061, 1006, and 1036 tons of oil eq. for titanium, PE/G-60/40, and PE/G-70/30, respectively.

### 3.3. Socioeconomic Aspect

Although the cost of the polymer-composite tubes for the equivalent level of heat-transfer rate as the Ti tubes is relatively conservative, it is still very promising for adapting polymer-composite-based evaporator tubes in MED plants, not only for the lower cost but also, more importantly, for localizing the design, manufacturing, maintenance, and replacement of the tubes. Local manufacturing of polymer composite tubes will significantly reduce, if not eliminate, transportation efforts, cost, and associated Ti-tube fabrication emissions. Localization of polymer-tube manufacturing will also give rise to creating a local manufacturing industry, leading to healthy economic diversification, generation of jobs, technological capacity, and talented human capital. Particularly for GCC, like Qatar, it will open an opportunity for efficient, local, and value-added use of petroleum-based polymers. This will lead to local materials to solve local problems and needs such as desalination and other piping and help establish a new line of value-added export in addition to their existing oil or natural gas shipments. Thus, economic diversification, local manufacturing industry, job creation, and capacity building will also give rise to the development of other second- and third-tier manufacturing and service businesses and trading. In the long run, improvements in health, education, and general human development index can be expected to be realized.

To illustrate the abovementioned socioeconomic impacts, it is possible to start with estimating the amount of local polymer composite tubing production needed for a typical size of the MED desalination plant. According to References [17,18], Rass Laffan, Qatar is a MED–TVC that consists of 10 evaporators that use 2700 tons of Ti tubes. The equivalent amount of polymer composite tubes is around 3600 tons. Based on the expected Qatar water demand growth of 7%, Qatar’s demand in the year 2050 will be twice the current water desalination plant capacity of 663 m^3^/year. Assuming the market will be evenly shared by the two competitive technologies RO and MED, the expected future MED project capacity of, accordingly, the required polymer composite tubes is 4.4 million tons. Assuming the polymer-composite tubes’ cost will be about $5000/ton, around $22 billion will be circulated within the local economy instead of continuing exporting Ti tubing and spending even more on maintenance and plant downtime. Development of manufacturing capacity, job creation, and potential research and development benefits are additional expected impacts but difficult to estimate.

## 4. Conclusions

In this work, the feasibility of a new thermally enhanced polymer based on our previous technical results of fabricating polymer–graphite composite material [15,16] was carried out against the commercially available titanium-tube-based MED evaporator. The HDPE-30% EG and HDPE-40% EG 40% performance is compared with titanium. The techno-economics analysis was performed for a commercial-scale MED plant with 2910 m^3^/day capacity, simulated using a VSP software developed by the author [36,37]. The energy and mass balance equations are solved, and the evaporator’s heat-transfer area, tube weight, and cost are calculated for both titanium- and polymer-composite tubes. A life-cycle assessment (LCA) was carried out, and the ReCiPe midpoint method was used to evaluate LCA indicators. In the end, a socioeconomic study was carried out, to evaluate the impacts of the polymer-tube material, and the outcomes of this study are as follows:

At the same evaporator capacity, we predicted the composite tube evaporator’s costs to be 36–52% and 27–45% lower than the titanium evaporator for the HDPE–EG 60/40 and HDPE/EG 70/30, respectively, assuming titanium tubes cost of $20,000/ton–$25,000/ton. Moreover, the composite tube evaporator cost will remain lower than the cost of the titanium evaporated, as long the composite tube remains below $9500/ton for HDPE-40% EG tubes and $13,500/ton for the HDPE-40% EG tubes. These results reveal the undoubted economic advantage of using polymer composites as the evaporator tubes’ material.

The LCA results indicate that the carbon emission can be reduced by 35% and 37% when HDPE-40% EG and HDPE-30% EG tubes are used instead of titanium tubes. The ozone depletion, acidification, and fossil fuel depletion for polymer tubes are comparable with titanium tubes. However, freshwater eutrophication caused by titanium tube production is lower than that of composite tube fabrication. The recycling of thermally enhanced polymers is not considered. However, after the end of life, reusing the material into other products would lower the overall environmental impacts.

Moreover, the polymer composite tubes can be produced locally, which will not only reduce the environmental impacts due to transportation but also create jobs for local manufacturing.

## Figures and Tables

**Figure 1 polymers-13-00681-f001:**
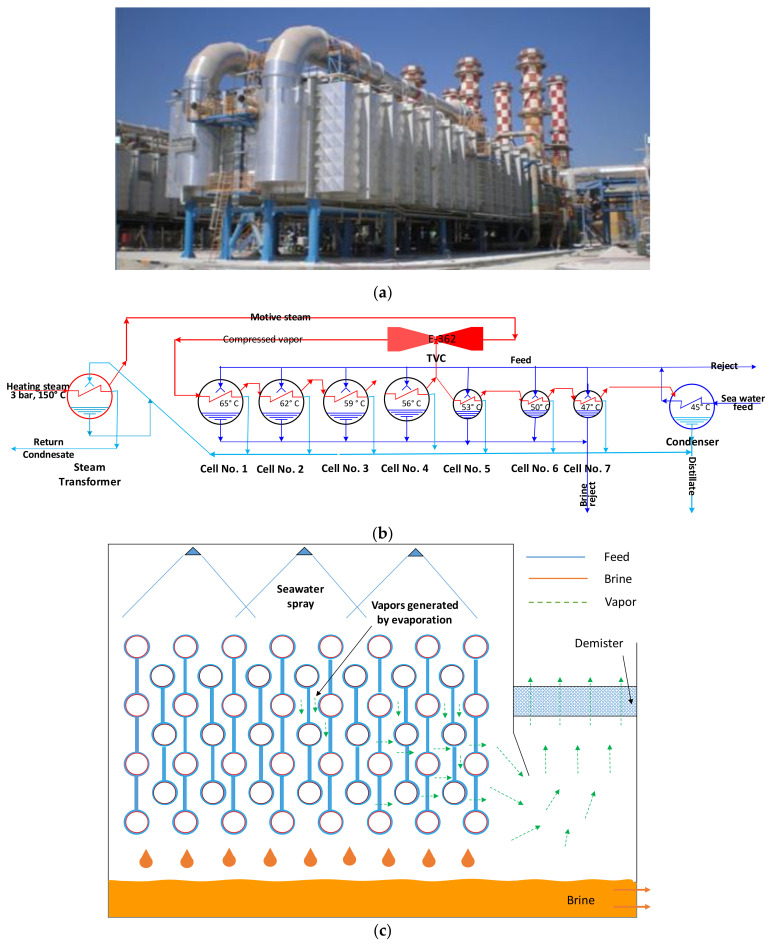
(**a**) Picture of multi-effect distillation (MED)/thermal vapor compressor (TVC) evaporator (6 MIGD (million imperial gallons per day)), Rass Laffan, Qatar; (**b**) process flow diagram of MED falling film evaporator; and (**c**) falling film evaporator schematic.

**Figure 2 polymers-13-00681-f002:**
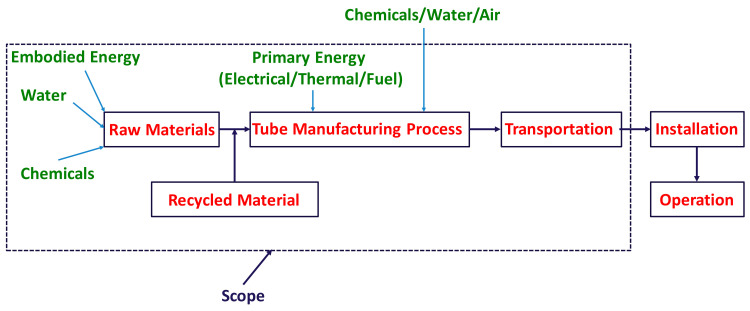
System boundary for tube manufacturing.

**Figure 3 polymers-13-00681-f003:**
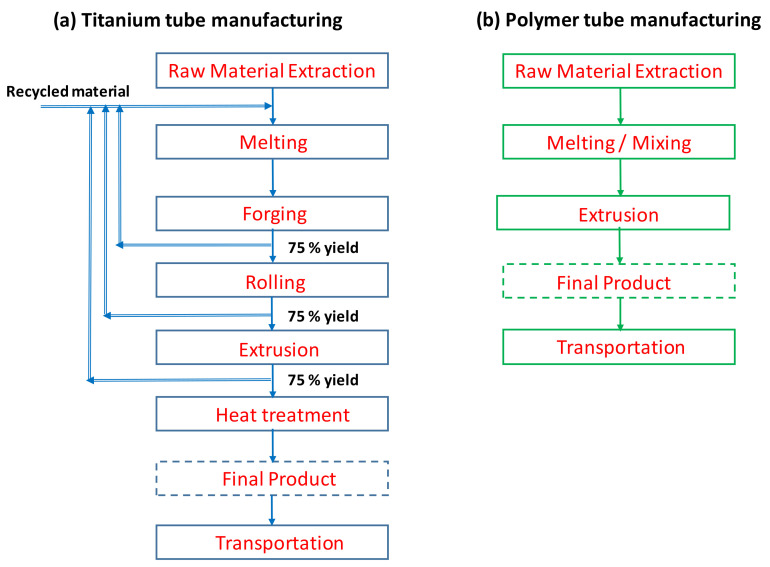
Tube manufacturing processes for (**a**) titanium and (**b**) thermally enhanced polymer.

**Figure 4 polymers-13-00681-f004:**
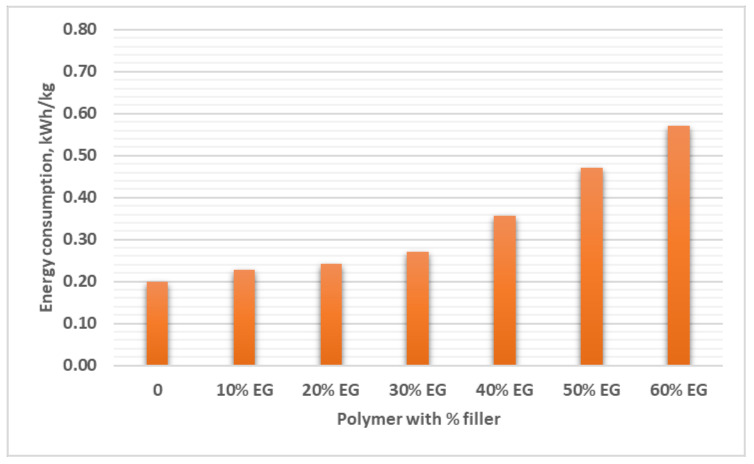
Energy consumption of the extrusion process.

**Figure 5 polymers-13-00681-f005:**
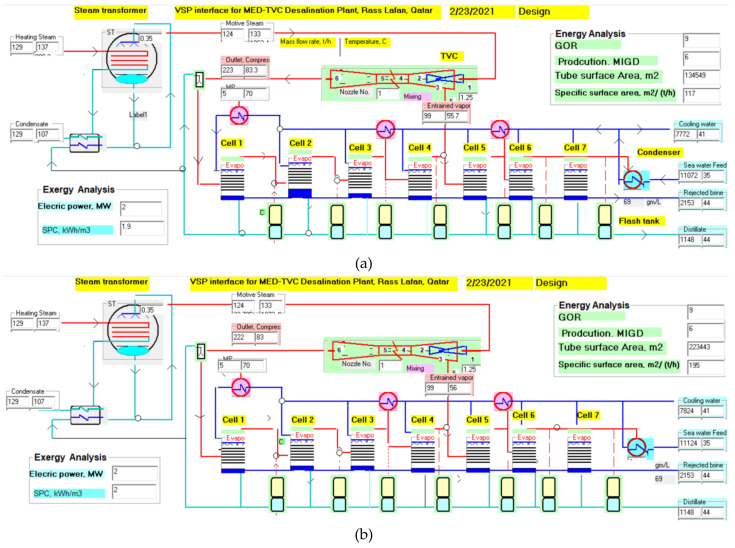
Interface of the visual simulation program (VSP) software for MED desalination plant (**a**) with titanium tubes and (**b**) with polymer-composite-material tube.

**Figure 6 polymers-13-00681-f006:**
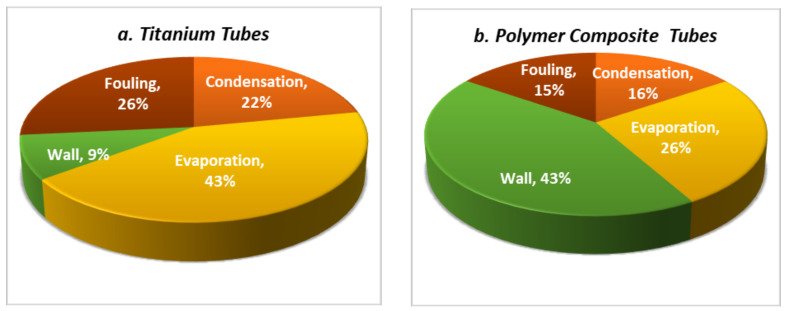
Thermal resistance distribution across the tube for (**a**) titanium and (**b**) polymer-G (60/40).

**Figure 7 polymers-13-00681-f007:**
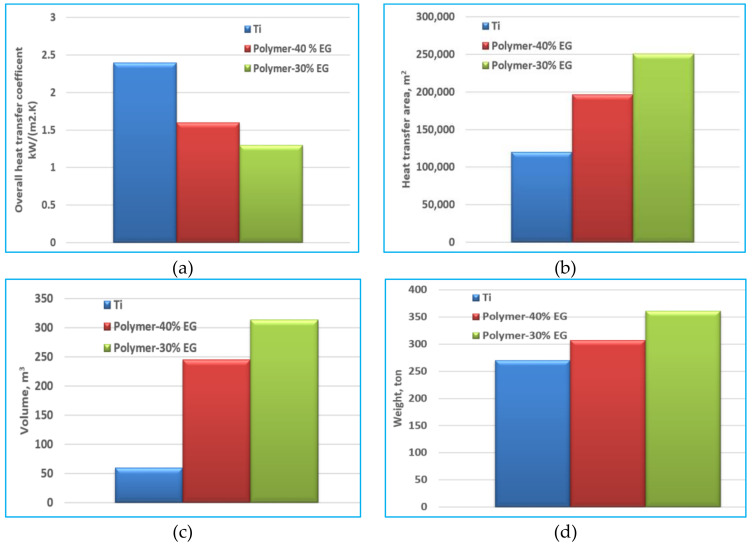
(**a**) Overall heat-transfer coefficient, (**b**) required heat-transfer area, (**c**) required material volume, and (**d**) required material weight for Ti, polymer-composite (60–40), and polymer-composite (70–30) MED evaporator.

**Figure 8 polymers-13-00681-f008:**
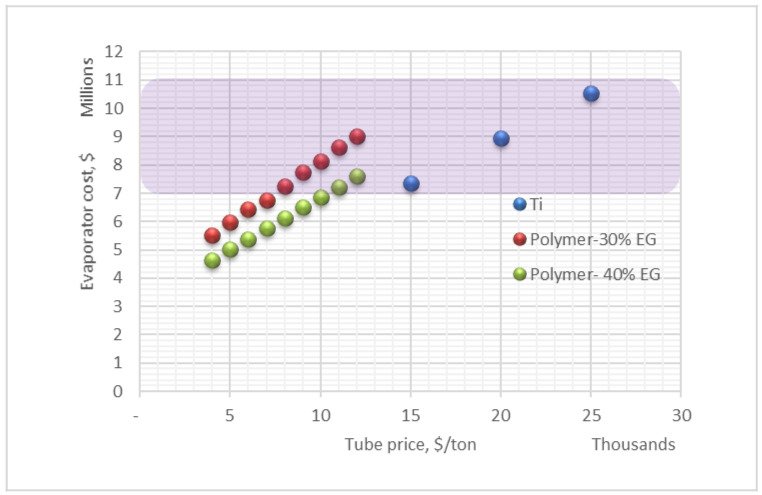
Evaporator cost (shell + tube) at different polymer-tube prices.

**Figure 9 polymers-13-00681-f009:**
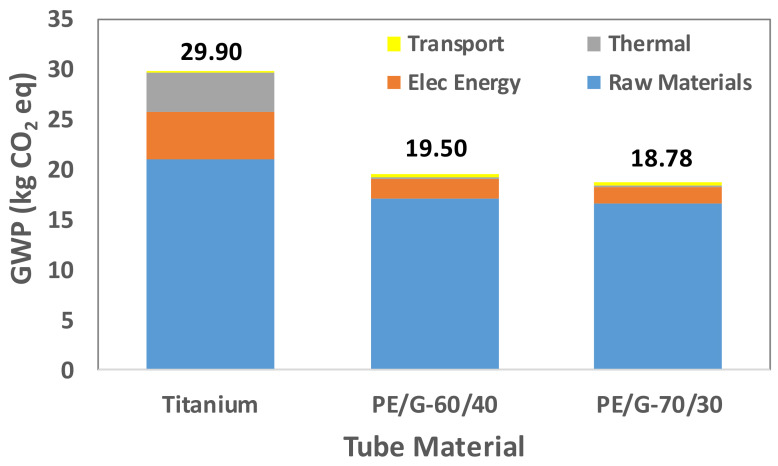
Global warming potential (GWP) for titanium and polymer tubes.

**Figure 10 polymers-13-00681-f010:**
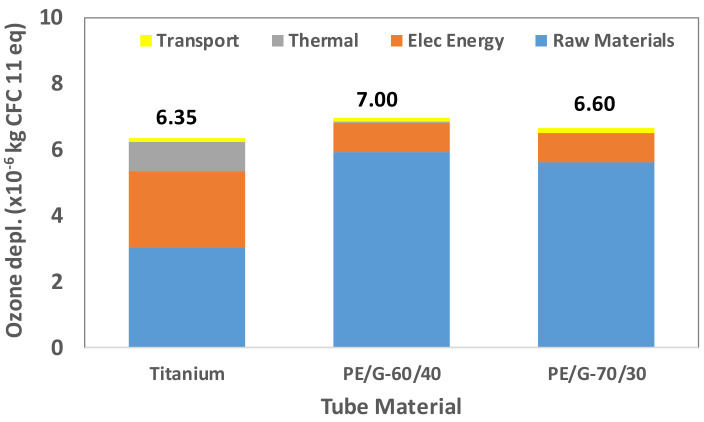
Ozone layer depletion (OD) for titanium and polymer tubes.

**Figure 11 polymers-13-00681-f011:**
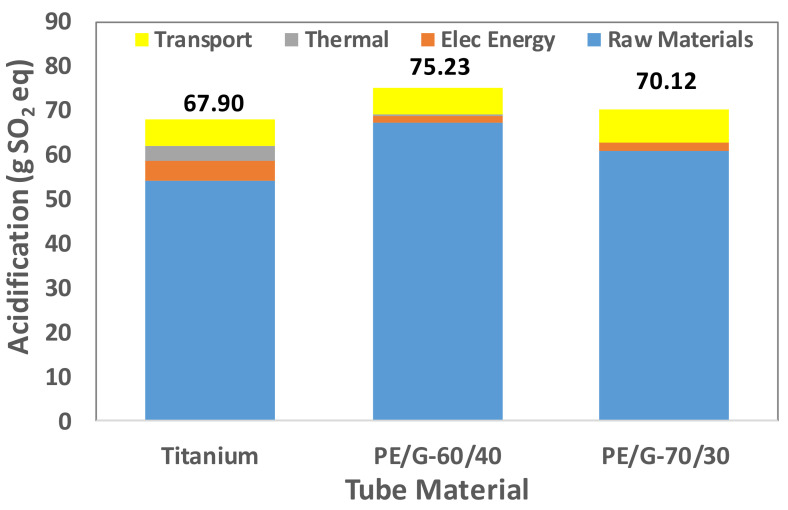
Acidification for titanium and polymer tubes.

**Figure 12 polymers-13-00681-f012:**
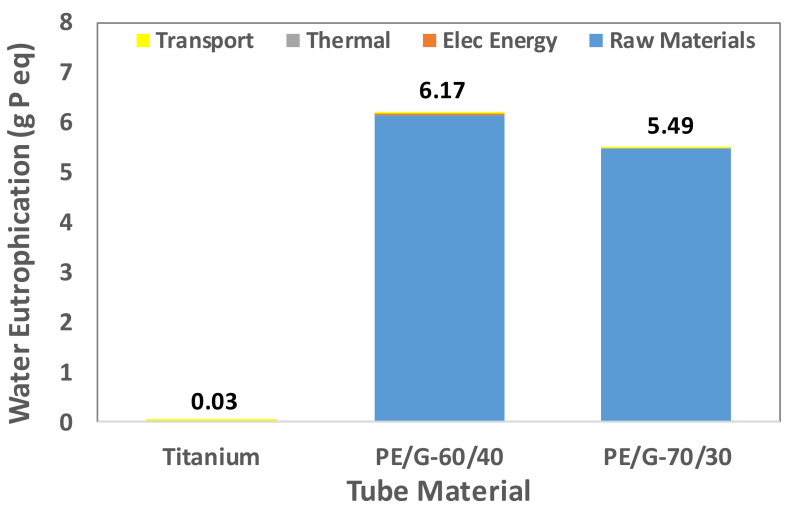
Freshwater eutrophication for titanium and polymer tubes.

**Figure 13 polymers-13-00681-f013:**
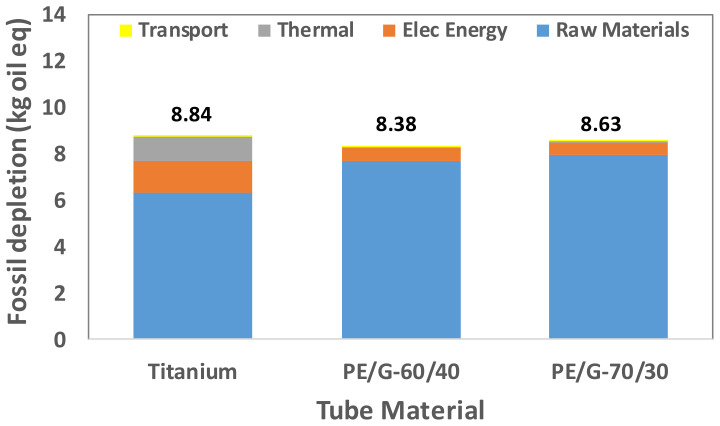
Fossil depletion for titanium and polymer tubes.

**Table 1 polymers-13-00681-t001:** Input modeling parameters for MED plant techno-economics.

	Ti	PE/EG 70/30	PE/EG 60/40
Plant capacity, ton/h	1148	1148	1148
Top brine temperature (TBT), C	65	65	65
Number of evaporators	7	7	7
Feed salinity, g/L	45	45	45
Rejected brine salinity, g/L	65	65	65
Tube length for effect 1, 2, 3, and 4, in m Steam	5	5	5
Tube diameter for effect 1, 2, 3, and 4, in mm	25.4	25.4	25.4
Tube length for effect 5, 6, and 7, in m	9	9	9
Tube diameter for effect 5, 6, and 7, in mm	38 mm	38 mm	38 mm
Wall thickness, mm	0.5	1.25	1.25
Density, kg/m^3^	4500	1150	1250
Thermal conductivity, W/m·K	17	3	5
Fouling factor, m^2^·K/kW	0.1	0.1	0.1

**Table 2 polymers-13-00681-t002:** Energy inputs for titanium-tube production.

	Process	Electrical Energy (kWh/kg)	Thermal Energy (kWh/kg)	Remarks
1	Ore to Ingot			Gabi
2	Forging/Billet	0.0885	1.23	[27]
3	Hot/Cold Rolling	0.2	0.6	730–815 °C [40]
4	Extrusion	0.128	0.43	980–1040 °C [40]
5	Heat Treatment (Annealing)	0.109	−	600 °C, Soaking 1 h [44]
6	Scrap melting to ingot	1.949	−	[45]

**Table 3 polymers-13-00681-t003:** Selected indicators categories for LCIA.

	Category	Abbreviation	Units
1	Global warming potential	GWP	kg CO_2_ eq.
2	Ozone depletion	OD	kg CFC-11 eq.
3	Acidification	AC	g SO_2_ eq.
4	Water eutrophication	WE	g P eq.
5	Fossil depletion	FD	Kg oil eq.

**Table 4 polymers-13-00681-t004:** The cost of the tubes fabricated using different filler loading.

Filler Percentage, %	0	10	20	30	40	50	60
Material cost, $/ton	1200	1780	2360	2940	3520	4100	4680
Mixing, $/ton		900	900	900	950	1000	1050
Extrusion, $/ton	400	400	400	400	430	470	500
Total cost, $/ton	1600	3080	3660	4240	4900	5570	6230

## Data Availability

The data presented in this study are available on request from the corresponding author.

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
