# Peer review of "Sustainability Assessment and Techno-Economic Analysis of Thermally Enhanced Polymer Tube for Multi-Effect Distillation (MED) Technology"

_polymers, 2021, doi:10.3390/polym13050681_

Round 1

Reviewer 1 Report

please see my comments attached. Please pay attention to the fact that you did not give a single detail about this 'new thermally enhanced polymer '. This term is highlighted in the title and is crucial for the whole paper.  If this information will be incorporated in the manuscript, it can be published. Overall, is s well written and well discussed paper.

Author Response

We considered all comments and revised them in the attached document and in some cases we added a paragraph in the methodology section (2.1) (in red color) and we amended 1 relevant reference [Ref. 16].  Hopefully, this amendment would clarify that this work is complementary work and parking on our previous work [ ref. 11,12, 15, and 16] and of course on the contribution of the open literature in the state of the art in terms of thermally enhanced polymer material.  To be more focus, in this article and due to the space limitation, we limited the scope to only perform a feasibility study based techno economics and LCA to answer some raised up questions such as:  

  • What is the feasible thermal concavity to produce reasonable overall capital cost reduction compared with a metallic tube such Ti?
  • What is the effect of specific weight (density) when compared with metal tubes?
  • What is the effect of thermal conductivity resistance compared to the convection heat transfer resistance between liquid and plate/tube surface?
  • What is the effect of thermal conductivity resistance compared to condensation/evaporation heat transfer resistance between vapor and plate/tube surface?
  • What is the effect of thermal conductivity compared to fouling on both plate surfaces?
  • Wall thickness effect compared with improving thermal conductivity?

All these factors have been considered in the VSP process simulation software to identify whether the improvement in thermal conductivity would be weighted by the reduction of the overall capital cost or Not?

Reviewer 2 Report

The manuscript named Sustainability assessment and techno economic analysis of thermally enhanced polymer tube for multi-effect distillation (med) technology deals with using of polyethylene/expanded graphite based thermally conductive composite and next, it describes socio-economic aspects if polymer composite tubes are used as an alternative in MED technology.

Here, let me introduce the main weakness and drawbacks of the presented article and the reasons why I cannot recommend it for publication in Polymers.

  1. The topic and research field of preparation thermally conductive polymers and its improvement is quite an established area and it is out of hot interest.
  2. This article does not bring any novelty either in the field of preparation of any novel material (composite based on polyethylene and expanded graphite are well known more than 15 years) or the technological or methodological field. Authors did not improve the methodology, they just used the established one.
  3. The prepared composite did not bring any breakthrough in the field of thermal conductivity. Presented results show that the thermal conductivity of prepared samples is similar or lower (3 and 5 W/m.K) in comparison with common commercially available products (in some cases more than 10 W/m.K). Moreover, authors did not introduce how they measured the thermal conductivity. It is Thermal conductivity flow, crossflow or through plane?
  4. The presented manuscript looks like case study more than research article.

Here I would write some points which downgrade the quality of the manuscript:

  • authors statement, that the environmental burden during the PE tubes preparation is higher than in case of Ti tubes, is fail and it is not supported by any reference.
  • English must be improved throughout whole text.
  • On Page 7: Missing references.
  • Page 11: Overall heat transfer coeff. is calculated. But there only values are introduced. Generally, in this manuscript is missing part about the experiments. How were the tubes prepared? Which conditions, which batch, and kind of polyethylene?

To summarize, the article is describing some engineering techniques for distillation of seawater, which is very important for countries close to Persian Gulf, but it does not bring any novelty and it is weak quality. From the reasons mentioned above, I recommend to reject.

Author Response

The manuscript named Sustainability assessment and techno-economic analysis of thermally enhanced polymer tube for multi-effect distillation (med) technology deals with using of polyethylene/expanded graphite based thermally conductive composite and next, it describes socio-economic aspects if polymer composite tubes are used as an alternative in MED technology.

Here, let me introduce the main weakness and drawbacks of the presented article and the reasons why I cannot recommend it for publication in Polymers.

  1. The topic and research field of preparation thermally conductive polymers and their improvement is quite an established area and it is out of hot interest.
  • Despite the advances in highly filled thermoplastic composites, materials formed using conductive filler exhibited limited improvement in thermal conductivity even at high filler concentration. In our ref. [12] we highlighted The limited improvement in the thermal conductivity owning to:
    1. Improvement of the thermal conductivity is only in the plane direction (x-direction) comparable to the y-direction (heat flow)
    2. The interfacial thermal resistance between the conductive filler and the surrounding polymer chains
    3. Adding filler at a large amount increases the viscosity of neat polymer which affect the processing (extrusion) Filler size effect still need optimization.
  1. This article does not bring any novelty either in the field of preparation of any novel material (composite based on polyethylene and expanded graphite are well known for more than 15 years) or the technological or methodological field. The authors did not improve the methodology, they just used the established one.
  • In this work and due to the space limitation we focused only on the feasibility study of using thermoplastic for thermal seawater desalination technique (MED) so that we limited the scope of work to address the cost analysis on our previous published work in the field as mentioned in ref. [ 11,12,14 and recent patented-pending idea as mentioned ref. 16]
  • I have added our new ref. [16] of a patented-pending idea in which we disclosed a new method to fabricate polymer-graphene/graphene composite to resolve the above mentioned technical issues and limitation of the current approaches of fabricating the large size thermoplastic.
  • [16] Abdelnasser Mabrouk, Chaudhry Usman, Mehamed, Ahmed Abdalla. Thermoplastic and method of manufacturing the same. D2020-057-01, US provision 63049803, (2020).
  1. The prepared composite did not bring any breakthrough in the field of thermal conductivity. Presented results show that the thermal conductivity of prepared samples is similar or lower (3 and 5 W/m.K) in comparison with common commercially available products (in some cases more than 10 W/m.K). Moreover, the authors did not introduce how they measured the thermal conductivity. It is Thermal conductivity flow, crossflow, or through-plane?
  • Would like to confirm that we limited the scope of this paper only to address the feasibility study which we believe is important for the decision-maker to decide GO/No-GO on the front edge before developing technology. Kindly, would like to referring to our recent publication [14] which presents our contribution and methodology of how to measure thermal conductivity.
  • Agree that there a massive work academic work that claimed a high thermal conductivity, however, there few who commercialized as mentioned in ref. [13] though the reported thermal conductivity in the order of 3-5.
  • In our current project and the patented methodology we could reach thermal conductivity 6 W/m.K at lower filler loading (25 % wt) and we are going to improve by increasing the load.
  1. The presented manuscript looks like a case study more than a research article.
  • It looks like a case study from the material development perspective, however from techno economics and process engineer perspective, it is a research topic in which we provided answers to   some raised up research questions such as:
  • What is the feasible thermal concavity to produce reasonable overall capital cost reduction compared with a metallic tube such Ti?
  • What is the effect of specific weight (density) when compared with metal tubes?
  • What is the effect of thermal conductivity resistance compared to the convection heat transfer resistance between liquid and plate/tube surface?
  • What is the effect of thermal conductivity resistance compared to condensation/evaporation heat transfer resistance between vapor and plate/tube surface?
  • What is the effect of thermal conductivity compared to fouling on both plate surfaces?
  • Wall thickness effect compared with improving thermal conductivity?

All these factors have been considered in the VSP process simulation software to identify whether the improve in thermal conductivity would be weighted by the  reduction of  the overall capital cost or Not?

Here I would write some points which downgrade the quality of the manuscript:

  • The author's statement, that the environmental burden during the PE tubes preparation is higher than in the case of Ti tubes, is fail and is not supported by any reference.

In terms of global warming potential, the polymer-based tubes are better than the titanium tubes (Figure 8). However, the other environmental parameters indicate that the titanium tubes are comparable with polymer-based tubes, or better in the case of water eutrophication (Figure 11). This is due to the following reasons:

  1. For thermally enhanced polymer tubes, for the thermal load of the heat exchanger, more weight and hence environmental burden.
  2. The graphite extraction and processing represent a major portion of the environmental burdens (please check [ref. 20] for further details). Expanded graphite indicates more than 70 % of the environmental burdens for polymer tube raw material.
  3. The viscosity of the thermally enhanced polymer is much higher than the plain polymer, which increases the extrusion energy by many folds [Please check Ref. No. 24].
  4. In a study by Ren et al. [34], it was concluded that the heat exchanger with steel tubes is better than the polyethylene tubes which were not thermally enhanced by incorporating high thermal conductive additives. This study shows that the environmental burden of polymer-based are significant and the incorporation of additives such as expanded graphite (which has higher environmental burden values) could lead to overall higher environmental impact.

We believe that our results are justified and supported by references as well based on the assumption of 40 % recycle of Ti tubes and 0.0 % of recycling of polymer –Graphite composite,

  • English must be improved throughout the whole text.
    • We have made improvements in the text for clarity.
  • On-Page 7: Missing references.
    • Revised
  • Page 11: Overall heat transfer coeff. is calculated. But there only values are introduced. Generally, this manuscript is missing the part about the experiments. How were the tubes prepared? Which conditions, which batch, and kind of polyethylene?

- Already answered these comments in the previous comments.

Round 2

Reviewer 1 Report

The authors made a tremendous effort during the revision process. The manuscript is greatly improved. Therefore, I suggest acceptance of the paper in its current form 

Author Response

I would like to thank Reviewer 1 for the kind  review and recommending for acceptance

Reviewer 2 Report

The authors improved the manuscript and highlighted that it deals with a feasibility study. 
Now I just recommend to intensify this fact in the last paragraph by adding of the word feasibility.

Author Response

We revised the documents as per Reviewer 2 and mentioned it is a feasibility study-based techno-economic, socioeconomic, and LCA analysis.